# Energy-Dependent Endocytosis Is Responsible for Skin Penetration of Formulations Based on a Combination of Indomethacin Nanoparticles and l-Menthol in Rat and Göttingen Minipig

**DOI:** 10.3390/ijms22105137

**Published:** 2021-05-12

**Authors:** Hiroko Otake, Mizuki Yamaguchi, Fumihiko Ogata, Saori Deguchi, Naoki Yamamoto, Hiroshi Sasaki, Naohito Kawasaki, Noriaki Nagai

**Affiliations:** 1Faculty of Pharmacy, Kindai University, 3-4-1 Kowakae, Higashi-Osaka, Osaka 577-8502, Japan; hotake@phar.kindai.ac.jp (H.O.); 2033420005s@kindai.ac.jp (M.Y.); ogata@phar.kindai.ac.jp (F.O.); 2045110002h@kindai.ac.jp (S.D.); kawasaki@phar.kindai.ac.jp (N.K.); 2Department of Ophthalmology, Kanazawa Medical University, 1-1 Daigaku, Uchinada, Kahoku, Ishikawa 920-0293, Japan; naokiy@kanazawa-med.ac.jp (N.Y.); mogu@kanazawa-med.ac.jp (H.S.)

**Keywords:** nanoparticle, endocytosis, transdermal delivery system, indomethacin, l-menthol

## Abstract

We previously designed a Carbopol gel formulation (N-IND/MEN) based on a combination of indomethacin solid nanoparticles (IND-NPs) and l-menthol, and we reported that the N-IND/MEN showed high transdermal penetration. However, the detailed mechanism for transdermal penetration of IND-NPs was not clearly defined. In this study, we investigated whether endocytosis in the skin tissue of rat and Göttingen minipig is related to the transdermal penetration of IND-NPs using pharmacological inhibitors of endocytosis. The pharmacological inhibitors used in this study are as follows: 54 µM nystatin, a caveolae-mediated endocytosis (CavME) inhibitor; 40 µM dynasore, a clathrin-mediated endocytosis (CME) inhibitor; and 2 µM rottlerin, a micropinocytosis (MP) inhibitor. The N-IND/MEN was prepared by a bead mill method, and the particle size of solid indomethacin was 79–216 nm. In both rat and Göttingen minipig skin, skin penetration of approximately 80% IND-NPs was limited by the stratum corneum (SC), although the penetration of SC was improved by the combination of l-menthol. On the other hand, the treatment of nystatin and dynasore decreased the transdermal penetration of indomethacin in rats and Göttingen minipigs treated with N-IND/MEN. Moreover, in addition to nystatin and dynasore, rottlerin attenuated the transdermal penetration of IND-NPs in the Göttingen minipigs’ skin. In conclusion, we found that l-menthol enhanced the SC penetration of IND-NPs. In addition, this study suggests that the SC-passed IND-NPs are absorbed into the skin tissue by energy-dependent endocytosis (CavME, CME, and/or MP pathways) on the epidermis under the SC, resulting in an enhancement in transdermal penetration of IND-NPs. These findings provide significant information for the design of nanomedicines in transdermal formulations.

## 1. Introduction

Oral drug delivery is a major route in the treatment of non-steroidal anti-inflammatory drugs (NSAIDs), although the oral administration of indomethacin (IND), which is an NSAID, has pre-systemic metabolism and undesirable side effects on the gastrointestinal tract. On the other hand, transdermal drug delivery (TDD) offers an ensured reduction in side effects in the gastrointestinal tract and the elimination of hepatic first-pass metabolism [1,2,3]. Moreover, the transdermal route provides sustained delivery [1,4] and permits pain-free and safe administration of drugs, resulting in an enhancement of compliance in patients [5]. Thus, the transdermal route is one of the methods that could overcome problems in the oral administration of NSAIDs [3]. However, transdermal penetration of drugs is lower than that in the mucous membrane of the digestive tube, and the stratum corneum (SC), which is the outermost layer, could be an obstacle for the design of transdermal formulation. Therefore, overcoming the SC, which presents a significant barrier, is important in the improvement of drug dermal permeability.

Many physical and chemical approaches for the enhancement of transdermal penetration have been reported to overcome this hurdle. Physical approaches referred to as ultrasound microneedle, microporation, magnetophoresis, electroporation, iontophoresis, and sonophoresis increase transdermal penetration and induce reversible disruption of the SC [6,7,8,9]. Chemical-permeation enhancers, pro-drugs, and colloidal formulations are referred to as chemical approaches and are used to enhance penetration of the skin by passive diffusion [6,7,8,9]. Furthermore, for many years, there has been a belief that nanoparticles (NPs) cannot penetrate intact skin, although it was reported that the optimal size of NPs (under 100 nm NPs) allowed for cellular uptake and permeated the SC [10,11,12,13,14]. In addition, the application of chemical penetration enhancers enhanced the cellular uptake of NPs [13,15]. From these findings, several nano-drug delivery systems (DDS), such as liposomes, solid NPs, polymeric NPs, dendrimers, lipid nanocarriers, and nanoemulsions, have rapidly evolved as the trend in TDD.

In our previous study, we also designed gels containing drug NPs, such as IND, ketoprofen, minoxidil, and raloxifene [10,11,13,14,16], and showed that drug NPs smaller than 100 nm can penetrate the SC [10,11,13,14,16]. Moreover, we found that a combination with l-menthol allowed drug NPs of approximately 100–200 nm in size to penetrate the skin [13]. On the other hand, the precise skin-penetrating mechanism of drug NPs is not clearly defined; however, our previous study, which used gel containing IND nanoparticles (IND-NPs) [13], showed that transdermal penetration of IND-NPs was attenuated under 4 °C condition (cold temperature incubation), which inhibits all energy-dependent uptake in cells [17]. In addition, many researchers have reported that drug NPs do not penetrate cells simply via diffusion and have studied the relationship between NP-based DDS and endocytosis [18,19,20,21,22]. Therefore, it is important to clarify the effect of energy-dependent uptake on transdermal penetration of IND-NPs with or without l-menthol to develop TDD based on NPs.

Generally, NPs smaller than 200 nm can be taken up by pinocytosis, and pinocytosis is mainly classified as caveolae-mediated endocytosis (CavME), clathrin-mediated endocytosis (CME), or macropinocytosis (MP) [23,24]. These individual cellular uptake pathways are specifically blocked by pharmacological inhibitors [25,26,27]. In this study, we investigated the effect of l-menthol on the SC penetration of IND-NPs using the skin of rats and Göttingen minipigs. In addition, we demonstrated that these forms of endocytosis (CavME, CME, and MP) are related to the skin penetration of IND from transdermal gel based on IND-NPs using pharmacological inhibitors (CavME-inhibitor nystatin, CME-inhibitor dynasore, and MP-inhibitor rottlerin) [25,26,27].

## 2. Results

### 2.1. Evaluation of Physical Properties in Transdermal Formulation Based on a Combination of IND-NPs and l-Menthol

Our previous study showed that 2-hydroxyproplyl-β-cyclodextrin (HPβCD) improved the aggregation of NPs [28] and that methylcellulose (MC) attained effective milling via treatment with a bead mill [28,29]. In addition, we found that the Carbopol gel was suitable for releasing drug NPs and that the combination with l-menthol made it possible for drug NPs to penetrate rat skin [10,11,13,14,16]. We prepared the Carbopol gel, incorporating the IND-NPs with (N-IND/MEN) or without l-menthol (N-IND) according to these previous studies [10,11,13,14,16], and evaluated the physical properties in the transdermal formulation based on NPs. Table 1 shows the particle size, number, zeta potential of NPs, viscosity, and solubility in the N-IND and N-IND/MEN. The IND particle size in the 1% N-IND was 103.1 nm (range 81–193 nm), with 22.6 × 10^10^ ± 1.2 × 10^10^ IND particles/0.3 g, and the zeta potential of solid IND was −21.1 mV. The solubility of IND was enhanced by bead-mill treatment, and the dissolved IND levels were 4.02-fold higher than those of the non-bead-mill treatment. On the other hand, the ratio of dissolved IND levels was low, with 1.5% of IND in the N-IND (more than 95.5% IND in the N-IND was solid IND). Moreover, the viscosity was 5.0 ± 0.1 Pa∙s. These characteristics in the formulations were similar regardless of whether l-menthol was added (Table 1).

### 2.2. Changes in Transdermal Penetration of IND-NPs in the Normal, SC-Removed, and Defrosted Skin

Figure 1 shows the IND transdermal penetration through the normal, SC-removed, and defrosted skin of rats in the N-IND and N-IND/MEN. The transdermal penetration of IND in the normal rat skin treated with N-IND increased linearly over time, the performance of IND penetration was significantly enhanced in the SC-removed skin, and the *AUC*_skin_ in SC-removed skin was 41.8 ± 3.9 µmol∙h/cm^2^. In addition, the combination with l-menthol also increased the transdermal penetration of IND, and there was no significant difference in the *AUC*_skin_ between normal skin treated with N-IND/MET (35.1 ± 1.3 µmol∙h/cm^2^) and SC-removed skin treated with N-IND. Moreover, the penetrated IND levels in the SC-removed skin treated with N-IND/MET were higher than those in the corresponding normal skin, although the rate of skin permeation in SC-removed skin/normal skin treated with N-IND/MEN was lower than that of N-IND. On the other hand, the IND penetration in the defrosted skin treated with N-IND or N-IND/MET was markedly lower in comparison with normal skin. Figure 2 shows the IND penetration of the normal, SC-removed, and defrosted skin of Göttingen minipigs in N-IND and N-IND/MEN. Similar to the result of the penetration study using rat skin, the combination with l-menthol enhanced transdermal penetration of IND in the normal skin, and IND penetration was significantly suppressed in the defrosted skin. Overall, the amount of IND penetration of the skin of Göttingen minipigs was lower than that of drug penetration in the skin of rats.

### 2.3. Effects of Energy-Dependent Endocytosis on Drug Skin Penetration of the Transdermal Formulation Based on A Combination of IND-NPs and l-Menthol

Figure 3 shows the changes in the transdermal penetration of IND from N-IND and N-IND/MEN in the rat skin treated with endocytosis inhibitors. Table 2 shows the penetration/permeability parameters analyzed from the data shown in Figure 3. No difference in IND transdermal penetration was observed in the control and two endocytosis inhibitor-treated groups (dynasore and rottlerin). The treatment of nystatin, however, slightly decreased IND transdermal penetration in N-IND, and *J*_c_ was also lower than that of the control. In contrast to the results for N-IND, both nystatin and dynasore attenuated IND transdermal penetration and *J*_c_ in the rat skin treated with N-IND/MEN. Figure 4 shows the changes in the transdermal penetration of IND from N-IND and N-IND/MEN in the Göttingen minipig skin treated with endocytosis inhibitors. Table 3 shows the penetration/permeability parameters analyzed from the data shown in Figure 4. There was no significant difference in the transdermal penetration and *J*_c_ in N-IND-applied Göttingen minipig skin treated with or without endocytosis inhibitors. On the other hand, treatment with nystatin, rottlerin, or dynasore significantly reduced the transdermal penetration of N-IND/MEN, and the *AUC*_skin_ values in nystatin, rottlerin, and dynasore were 57.1%, 70.7%, and 57.7% of the control, respectively.

## 3. Discussion

IND is an NSAID that directs anti-inflammatory action through the inhibition of COX-2 enzymes. IND is often applied as a TDD method to reduce side effects, such as in the gastrointestinal tract. We previously designed gels containing IND-NPs and l-menthol and reported that the IND-NPs released from transdermal formulation (gel) showed deep skin penetration [13]. However, the detailed mechanism for skin penetration of NPs is not clearly defined. Therefore, in this study, we demonstrated whether these forms of endocytosis are related to the skin penetration of IND from transdermal formulations based on IND-NPs using pharmacological inhibitors of CavME, CME, and MP [25,26,27].

First, we prepared the N-IND and N-IND/MEN and measured the particle size, number, zeta potential, viscosity, and solubility in N-IND and N-IND/MEN (Table 1). The solid IND in N-IND and N-IND/MEN showed that at nanosizes and for the combination with l-menthol, the characteristics of IND transdermal formulations were not affected. Next, we attempted to clarify the detailed mechanism of transdermal penetration in N-IND and N-IND/MEN. The skin was mainly classified as epidermis, dermis, and subcutaneous tissue. The SC, the top layer of the epidermis, consists of keratin-filled corneocytes and works as a barrier, resulting in limited drug permeability through the skin. In general, rat skin is widely used in the study for permeability of drugs through the skin. On the other hand, the structure and penetration of minipig skin are similar to those of human skin in comparison with rat skin and better reflect the permeation characteristics in comparison with rat skin [30,31,32,33]. In particular, the Göttingen minipig is a particularly good skin model for in vitro penetration of human skin. The Göttingen minipigs were also used in experiments for in vivo pharmacokinetic studies of dermal drug products and predicted human pharmacokinetics profiles [30,31,32,33]. From these previous reports, we used both rat and Göttingen minipigs as the animal skin model and demonstrated how l-menthol increases the skin penetration of IND-NPs in the skin of rats and Göttingen minipigs.

Our previous studies [10,11,13] showed that the rapid increase (burst) in drug penetration was observed when the skin tissue was broken in the rat skin penetration experiments, and it was reported that the integrity of rat skin tissue is preserved for 24 h in the skin penetration experiments [10,11,13]. Taken together, we determined the measurement time at 24 h, and the rapid increase (burst) in IND penetration was not observed in the study (Figure 1, Figure 2, Figure 3 and Figure 4). The transdermal penetration of IND-NPs was enhanced by the combination with l-menthol in both rats and Göttingen minipigs (Figure 1 and Figure 2). The results support our previous reports using IND, ketoprofen, minoxidil, and raloxifene [10,11,13,14,16]. The SC acts as a barrier for the skin and prevents the penetration of drugs. Therefore, we investigated the relationship between SC and transdermal penetration of IND-NPs. The *AUC*_skin_ in SC-removed skin treated with N-IND was 4.30- and 4.97-fold that of normal skin of rats and Göttingen minipigs, respectively (Figure 1 and Figure 2). Moreover, no significant difference was observed between *AUC*_skin_ in SC-removed skin treated with N-IND and N-IND/MET in both rats and Göttingen minipigs (Figure 1 and Figure 2). It was reported that l-menthol is an enhancer of drug skin permeation, since l-menthol enhances skin penetration by altering the barrier properties of the SC [34]. Taken together, the skin penetration of IND-NPs of 81–193 nm is suggested to be severely limited by the SC. Moreover, the combination with l-menthol may alter the barrier properties of the SC and may enhance the permitted penetration of IND-NPs.

Furthermore, we demonstrated the transport pathways of IND-NPs into skin. Our previous study showed that the transdermal penetration of IND-NPs was prevented under cold temperature (4 °C), inhibiting all energy-dependent uptake [17]. In this study, we investigated the changes in transdermal penetration of IND-NPs using defrosted skin, which is non-survival skin. In the skin of rats and Göttingen minipigs, the transdermal penetration of IND-NPs in the defrosted skin was attenuated in comparison with that in normal skin. Furthermore, the *AUC*_skin_ in defrosted skin treated with N-IND/MET was 0.83- and 0.26-fold that of the corresponding normal skin treated with N-IND in the rats and Göttingen minipigs, respectively (Figure 1 and Figure 2). In addition, the *AUC*_skin_ in defrosted skin without SC treated N-IND/MET (*AUC*_skin_ in rats, 14.3 ± 1.8; *AUC*_skin_ in Göttingen minipigs, 6.7 ± 0.9 mmol∙h/cm^2^, *n* = 4) was also lower than SC-removed skin treated with N-IND/MET. These results suggested that the survival of cells in the epidermis excluding SC and dermis was related to the promotion of the transdermal absorption of IND-NPs.

Recent reports have shown that NPs can penetrate deep into the skin depending on their surface charge, size, and the material of the NPs [35]. Moreover, it was reported that the size and shape of the NPs are very important factors in the permeation pathways via endocytosis [36,37]. In addition, many researchers demonstrated that not only do the NPs enter cells by diffusion, but the penetration of NPs is directed by the various endocytosis pathways, such as CavME, CME, and MP [18,19,20,21,22,38]. CavME is often observed in viruses and bacteria (pathogens) to avoid lysosomal degradation [39]. CavME is a common pathway for cellular entry, covering a broad range of particles of sizes 60–80 nm [30] and bypassing lysosomes [40]. CME is also an important pathway for cellular entry and is taken up by approximately 100 nm vesicles [23]. The MP is an actin-regulated pathway and is exploited by particles 100 nm to 5 µm in size [41,42,43,44]. From these findings, we investigated whether this endocytosis (CavME, CME, and MP) is related to the transdermal penetration of IND-NPs of approximately 80–200 nm in size in this study. Endocytic inhibitors are drugs that block individual cellular uptake pathways [25,26,27], and nystatin, dynasore, and rottlerin are specific inhibitors for CavME, CME, and MP, respectively [25,26,27]. These inhibitors were used to prevent energy-dependent endocytosis in this study. In the skin treated with N-IND, there were no significant differences in the *AUC*_skin_ of the group treated with these endocytic inhibitors (Figure 3 and Figure 4). On the other hand, the *AUC*_skin_ of the group treated with nystatin and dynasore was attenuated in both rats and Göttingen minipigs treated with N-IND/MEN. Moreover, the *AUC*_skin_ in the Göttingen minipigs treated with N-IND/MEN was also decreased by the treatment of rottlerin, which suggests that the SC-passed IND-NPs were taken up by energy-dependent endocytosis treated with N-IND/MEN (Figure 3 and Figure 4), resulting in an enhancement of the transdermal penetration of IND-NPs. In addition, multi-endocytosis, such as CavME, CME, and/or MP, may be activated in the epidermis excluding SC and dermis. The particle size of IND-NPs in this study was approximately 80–200 nm, and it is known that the differences in the sizes and shapes of the NPs are related to the induction of CavME, CME, and MP. Therefore, the difference in size was suggested to lead to multi endocytosis and the contribution to the route via different endocytosis pathways was suggested to affect the transdermal penetration of IND-NPs. On the other hand, the related endocytosis was different in the animal skin model, since rottlerin did not suppress the skin penetration of IND-NPs in rat skin but attenuated the skin penetration in Göttingen minipigs. Further studies are needed to clarify the differences between rats and Göttingen minipigs. In addition, evaluating the relationships between the in vitro and in vivo pharmacokinetics studies of IND-NPs is important. Therefore, we plan to analyze in vivo rat skin penetration and histopathological tissue.

## 4. Materials and Methods

### 4.1. Animals

Male, six-week-old Wistar rats weighing 281 ± 8 g (mean ± S.E.M, *n* = 82) were provided by Kiwa Laboratory Animals Co., Ltd. (Wakayama, Japan), and the skin of male Göttingen minipigs, age five months, was purchased from Oriental Yeast Co., Ltd. (Tokyo, Japan). The rats were provided with a water and a CE-2 formulation diet (Clea Japan Inc., Tokyo, Japan) and housed under normal conditions. The skin of rats and Göttingen minipigs was obtained from the abdomen (normal skin), and the SC was removed using the tape-stripping method (SC-removed skin). For the defrosted skin, the removed skin was stored at −30 °C and defrosted in preparation for the experiment. The experiments using animals were approved by the animal care and user committee of Kindai University and carried out in accordance with the Pharmacy Committee Guidelines (project identification code KAPS-25-002, 1 April 2013 and KAPS-31-011, 1 April 2019).

### 4.2. Chemicals

All reagents used were the highest purity commercially available. Briefly, the HPβCD was provided by Nihon Shokuhin Kako Co., Ltd. (Tokyo, Japan), and MC was obtained from Shin-Etsu Cheical Co., Ltd. (Tokyo, Japan). IND powder, l-menthol, isoflurane, cytochalasin D, 4-(2-hydroxyethyl)-1-piperazineethanesulfonic acid (HEPES) buffer, and propyl p-hydroxybenzoatewere were purchased from Wako Pure Chemical Industries, Ltd. (Osaka, Japan). Rottlerin (micropinocytosis inhibitor) and dynasore (CME inhibitor) were obtained from Nacalai Tesque (Kyoto, Japan). Carbopol (carboxypolymethylene, Carbopol^®^ 934) and nystatin were purchased from Serva (Heidelberg, Germany) and Sigma–Aldrich Japan (Tokyo), respectively.

### 4.3. Design of the Transdermal Formulation Based on IND-NPs

The transdermal formulation based on IND-NPs was prepared according to our previous study [10,11,13,14,16]. A total of 1% IND and 0.5% MC were added to a 5% HPβCD solution and stirred for 5 h at room temperature (22 °C), and the dispersions were transferred to a 1.5 mL tube with zilconia beads (diameter, 100 µm) and milled by the Micro Smash MS-100R at 5500 rpm for 30 s for a total of 30 times at 4 °C (Tomy Digital Biology Co., Ltd., Tokyo, Japan). Furthermore, the milled dispersions were treated at 1500 rpm for 3 h with a Shake Master NEO BMS-M10N21 (Bio-Medical Science Co., Ltd., Tokyo, Japan) and the IND-NPs. After that, the IND-NPs were incorporated into the 3% Carbopol gel with (N-IND/MEN) or without 2% l-menthol (N-IND). The compositions of N-IND and N-IND/MEN were as follows: 1% IND, 0.5% MC, 5% HPβCD, 3% Carbopol, and/or 2% l-menthol. In the preliminary research, we confirmed that most of the IND-NPs in the N-IND and N-IND/MEN were released from the gels [13].

### 4.4. Characteristics of Transdermal Formulations Based on IND-NPs

The characteristics of the transdermal formulation based on IND-NPs were determined according to our previous reports [10,11,13,14,16]. Briefly, the zeta potential of IND-NPs was measured using a model 502 zeta-potential analyzer (Nihon Rufuto Co., Ltd., Tokyo, Japan). The particle-size distribution and number of IND-NPs were determined by a NANOSIGHT LM10 (QuantumDesign Japan, Tokyo, Japan), and the measurement conditions were as follows: time 60 s, viscosity of the suspension 1.27 mPa∙s, and wavelength 405 nm. The viscosity of the IND transdermal formulations was measured at 22 °C by a Brookfield digital viscometer (Brookfield Engineering Laboratories, Inc., Middleboro, MA, USA). The soluble IND and solid IND (IND-NPs) in the transdermal formulations were separated by centrifugation (1 × 10^5^ g) using an Optima^TM^ MAX-XP Ultracentrifuge (Beckman Coulter, Osaka, Japan), and the levels of soluble IND and IND-NPs were analyzed by the HPLC method described below.

### 4.5. Measurement of IND by HPLC Method

The IND was measured according to a previous study [13,28]. The IND levels were analyzed on an HPLC LC-20AT system at 35 °C (Shimadzu Corp., Kyoto, Japan). The mobile phase was acetonitrile/50 mM acetic acid (40/60, *v*/*v*) at a flow rate of 250 µL/min, and an Inertsil^®^ ODS-3 column (GL Science Co., Inc., Tokyo, Japan) was used. A propyl p-hydroxybenzoate was used as an internal standard, and the wavelength at detection was determined as 254 nm. The calibration curve was linear (y = 0.2668x − 0.0026, r = 0.9999), and the detection limit was approximately 0.1 µg/mL.

### 4.6. In Vitro Skin Penetration of Transdermal Formulation Based on IND-NPs

The in vitro transdermal penetrations of N-IND and N-IND/MEN were performed according to a previous study using the Franz diffusion cell [10,11,13,14]. The skins of rats and Göttingen minipigs were set as the Franz diffusion cell, and the O-ring flange (1.6 cm i.d.) was attached onto the skin. Then, 0.3 g of 1% N-IND and N-IND/MEN was applied to the O-ring flange. The reservoir chamber (dermal side) was soaked in a buffer (0.85% NaCl-10 mM phosphate buffer, pH 7.4) at 37 °C, and 100 µL solution was withdrawn from the dermal side as samples. The IND in the samples was extracted by methanol, and the IND levels were analyzed by the HPLC method described above. For the analysis of the endocytosis pathways, the endocytosis inhibitors and its concentration were determined according to previous studies [25,26,27], and CavME-inhibitor nystatin (54 µM in 0.5% dimethylsulfoxide) [26], CME-inhibitor dynasore (40 µM in 0.5% dimethylsulfoxide) [25], and MP-inhibitor rottlerin (2 µM in 0.5% dimethylsulfoxide) [27] were pretreated for the removed skin of rats and Göttingen minipigs for 1 h prior to the treatment of the transdermal formulation. Moreover, the reservoir chamber (dermal side) was soaked in a buffer (0.85% NaCl–10 mM phosphate buffer, pH 7.4) with an endocytosis inhibitor at 37 °C. The areas under the penetrated IND concentration–time curves on the dermal side (*AUC*_Skin_) were analyzed according to the trapezoidal rule up to the IND levels at 24 h (last measurement point). In addition, the penetration rate (*J*_c_), penetration coefficient through the skin (*K*_p_), skin/preparation partition coefficient (*K*_m_), lag time (*t*_lag_), and diffusion constant within the skin (*D*) were calculated according to Equations (1)–(3) [11,14]:(1)$$t_{\mathrm{lag}}=\frac{\delta^{2}}{6D}$$
(2)$$J_{c}=\frac{K_{m}\cdot D\cdot J_{\mathrm{drug}}}{\delta }=K_{p}\cdot C_{\mathrm{drug}}$$
(3)$$Q_{t}=J_{c}\cdot A(t-t_{\mathrm{lag}})$$
where *Q*_t_ is the effective area of the skin and *A* is 2 cm^2^. The thicknesses of the skins of rats and Göttingen minipigs (*δ*) were 0.086 ± 0.011 cm and 0.133 ± 0.021 cm, respectively (mean of 10 independent skin samples, mean ± S.E.M)); *C*_drug_ represents IND levels on the dermal side at time *t*.

### 4.7. Statistical Analysis

The data are presented as the mean ± S.E.M and were analyzed by one-way analysis of variance (ANOVA) followed by Dunnett’s multiple comparison. Statistical significance was set at *p* < 0.05.

## 5. Conclusions

We found that a combination with l-menthol enhanced the SC penetration of IND-NPs and that energy-dependent endocytosis was related to the high transdermal penetration of gels based on NPs. We hypothesized that the skin penetration of IND-NPs 81–193 nm in size was severely limited by the SC, but l-menthol altered the barrier properties of the SC and permitted SC penetration of IND-NPs. The SC-passed IND-NPs were taken up into the skin tissue by energy-dependent endocytosis (CavME, CME, and/or MP pathways) on the epidermis, excluding SC, resulting in enhanced skin penetration of IND-NPs (Figure 5). These important findings can be used to develop further studies aimed at designing TDD methods based on nanomaterials.

## Figures and Tables

**Figure 1 ijms-22-05137-f001:**
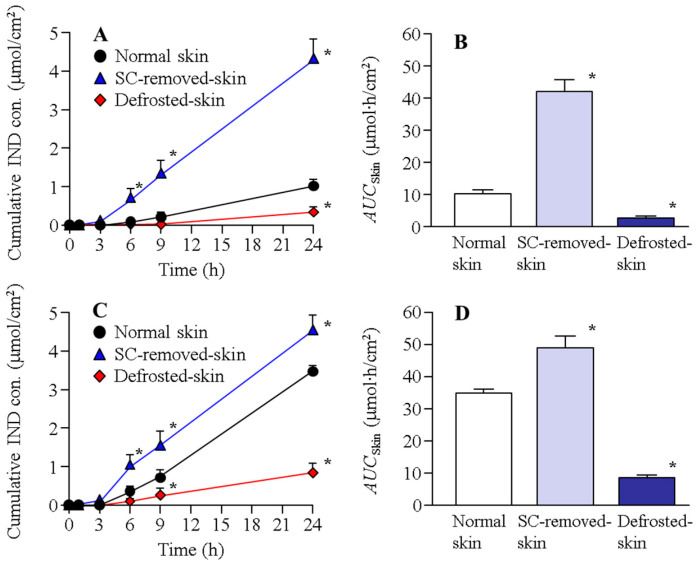
Transdermal penetration of indomethacin nanoparticles (IND-NPs) in normal, SC-removed, and defrosted skin of rats. (**A**,**B**) Penetration profile (**A**) and *AUC*_Skin_ (**B**) in skin of rats treated with N-IND. (**C**,**D**) Penetration profile (**C**) and *AUC*_Skin_ (**D**) in skin of rats treated with N-IND/MEN. The SC was removed by tape stripping (SC-removed skin), and the frozen skin stored at −30 °C was used as defrosted skin. The IND transdermal formulation was applied to normal (with SC), SC-removed (without SC), and defrosted-skin. Mean ± S.E.M. *n* = 7. * *p* < 0.05 vs. normal skin for each category. Although the SC attenuated the transdermal penetration of IND-NPs in N-IND, the SC penetration was enhanced by the combination with l-menthol. In the defrosted skin, the transdermal penetration of IND-NPs with or without l-menthol was lower than that in the corresponding non-defrosted skin.

**Figure 2 ijms-22-05137-f002:**
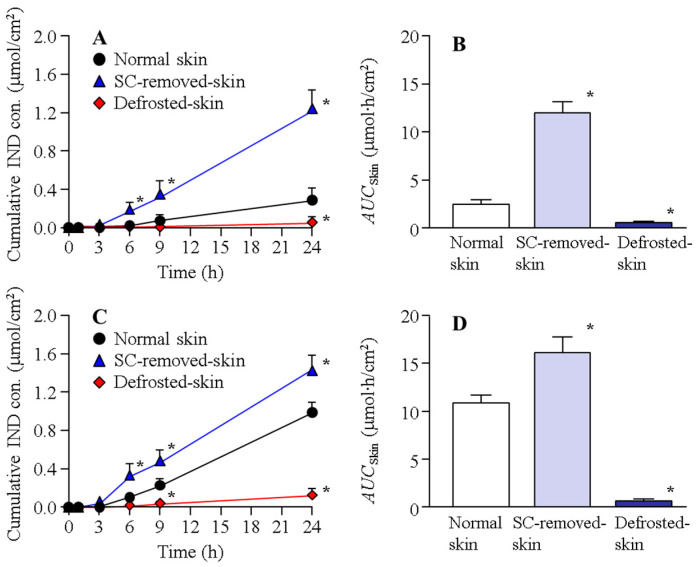
Transdermal penetration of IND-NPs in normal, SC-removed, and defrosted skin of Göttingen minipigs. (**A**,**B**) Penetration profile (**A**) and *AUC*_Skin_ (**B**) for skin of Göttingen minipigs treated with N-IND. (**C**,**D**) Penetration profile (**C**) and *AUC*_Skin_ (**D**) for skin of Göttingen minipigs treated with N-IND/MEN. The SC was removed by tape stripping (SC-removed skin), and the frozen skin at −30 °C was used as defrosted skin. The IND transdermal formulations were applied to normal (with SC), SC-removed (without SC), and defrosted skin. Mean ± S.E.M. *n* = 7–8. * *p* < 0.05 vs. normal skin for each category. The combination with l-menthol improved the SC penetration of IND-NPs. The penetrations of IND-NPs in both N-IND and N-IND/MEN were significantly lower in the defrosted skin than in the normal skin.

**Figure 3 ijms-22-05137-f003:**
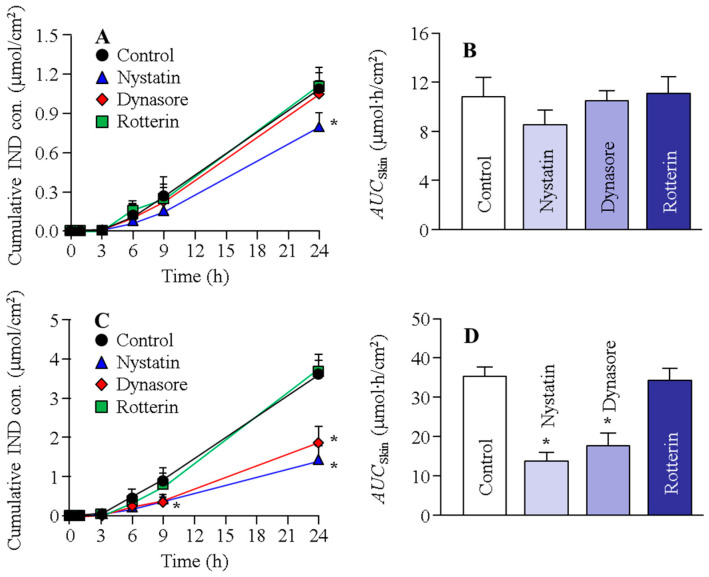
Effect of endocytosis on the transdermal penetration of IND released from N-IND and N-IND/MEN in the rat skin. (**A**,**B**) Penetration profile (**A**) and *AUC*_Skin_ (**B**) in the skin of rats treated with N-IND. (**C**,**D**) Penetration profile (**C**) and *AUC*_Skin_ (**D**) in the skin of rats treated N-IND/MEN. The skin was treated with endocytosis inhibitors (nystatin, dynasore, or rottlerin). Mean ± S.E.M. *n* = 5–8. * *p* < 0.05 vs. control for each category. The transdermal penetration of IND in the N-IND was slightly decreased by nystatin. Both nystatin and dynasore attenuated the transdermal penetration of IND in N-IND/MEN.

**Figure 4 ijms-22-05137-f004:**
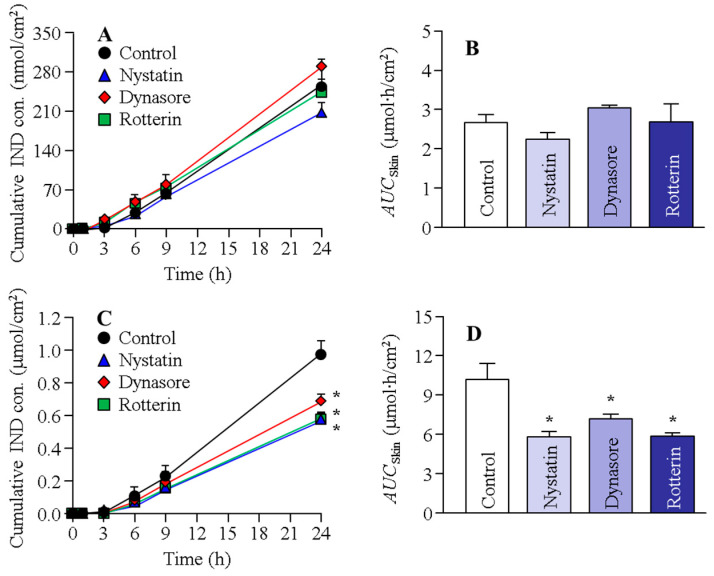
Effect of endocytosis on the transdermal penetration of IND released from N-IND and N-IND/MEN in the Göttingen minipig skin. Penetration profile (**A**) and *AUC*_Skin_ (**B**) in the skin of Göttingen minipigs treated with N-IND. Penetration profile (**C**) and *AUC*_Skin_ (**D**) in the skin of Göttingen minipigs treated with N-IND/MEN. The skin was treated with endocytosis inhibitors (nystatin, dynasore, or rottlerin). Mean ± S.E.M. *n* = 7–10. * *p* < 0.05 vs. control for each category. The nystatin tended to decrease the penetration of IND in N-IND. The penetration of IND in N-IND/MEN was significantly attenuated by the treatment of nystatin, dynasore, and rottlerin.

**Figure 5 ijms-22-05137-f005:**
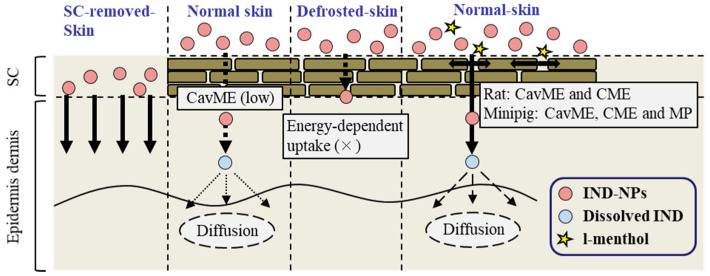
Mechanism for the skin-penetration process from transdermal formulations based on the combination of IND-NPs and l-menthol. × shows no-detectable. The size of dashed arrow represent the penetration level.

**Table 1 ijms-22-05137-t001:** Characteristics of the IND transdermal formulations.

Formulation	Particle Size nm (Range)	Number of NPs × 10^10^ Particles/0.3 g	Zeta Potentials mV	Viscosity Pa∙s	Solubility µM
N-IND	103.1 (81–193)	22.6 ± 1.2	−21.1 ± 0.7	5.0 ± 0.1	391.2 ± 10.3
N-IND/MEN	106.3 (79–216)	22.1 ± 1.4	−19.9 ± 0.6	5.0 ± 0.1	408.5 ± 11.7

Measurement of the experiments was performed at 22 °C. Mean ± standard error of the mean (S.E.M). *n* = 8.

**Table 2 ijms-22-05137-t002:** Penetration/permeability parameters of IND transdermal formulations in the in vitro penetration of rat skin.

Treatment	*J*_c_ (nmol/cm^2^/h)	*K*_p_ (×10^−4^ cm/h)	*K*_m_ (×10^−2^)	*τ* (h)	*D* (×10^−4^ cm^2^/h)
N-IND Control	43.6 ± 4.3	1.55 ± 0.19	2.76 ± 0.30	2.09 ± 0.08	4.04 ± 0.11
Nystatin	34.9 ± 3.2 *	1.23 ± 0.15 *	2.09 ± 0.18 *	1.90 ± 0.05	4.43 ± 0.10 *
Dynasore	43.2 ± 4.8	1.54 ± 0.23	2.72 ± 0.35	2.08 ± 0.09	4.07 ± 0.13
Rottlerin	43.7 ± 4.1	1.52 ± 0.18	2.69 ± 0.29	2.08 ± 0.08	4.09 ± 0.11
N-IND/MEN Control	158.7 ± 5.7	5.69 ± 0.19	11.0 ± 0.33	2.31 ± 0.04	3.78 ± 0.13
Nystatin	53.6 ± 0.38 ^#^	1.69 ± 0.18 ^#^	3.15 ± 0.29 ^#^	2.10 ± 0.07 ^#^	4.02 ± 0.12
Dynasore	61.2 ± 0.43 ^#^	1.78 ± 0.19 ^#^	3.86 ± 0.38 ^#^	2.13 ± 0.05 ^#^	3.95 ± 0.13
Rottlerin	159.1 ± 5.9	5.71 ± 0.20	11.3 ± 0.31	2.30 ± 0.04	3.80 ± 0.15

Mean ± S.E.M. *n* = 5–8. * *p* < 0.05 vs. control in N-IND for each category. ^#^ *p* < 0.05 vs. control in N-IND/MEN for each category.

**Table 3 ijms-22-05137-t003:** Penetration/permeability parameters of IND transdermal formulations in the in vitro penetration of Göttingen minipig skin.

Treatment	*J*_c_ (nmol/cm^2^/h)	*K*_p_ (×10^−4^ cm/h)	*K*_m_ (×10^−2^)	*τ* (h)	*D* (×10^−4^ cm^2^/h)
N-IND Control	11.7 ± 1.14	4.16 ± 0.39	5.10 ± 0.50	1.46 ± 0.08	5.78 ± 0.33
Nystatin	8.44 ± 0.41	3.03 ± 0.13 *	3.48 ± 0.06 *	1.36 ± 0.09	6.15 ± 0.39
Dynasore	12.9 ± 0.31	4.59 ± 0.11	4.57 ± 0.34	1.20 ± 0.06	7.09 ± 0.41
Rottlerin	11.4 ± 1.70	4.01 ± 0.59	4.26 ± 0.50	1.27 ± 0.15	6.63 ± 0.93
N-IND/MEN Control	44.4 ± 3.39	15.9 ± 1.55	20.7 ± 1.39	1.56 ± 0.13	5.46 ± 0.41
Nystatin	26.1 ± 1.45 ^#^	9.3 ± 0.50 ^#^	13.3 ± 0.58 ^#^	1.67 ± 0.04	5.10 ± 0.11
Dynasore	30.6 ± 2.03 ^#^	10.9 ± 0.74 ^#^	14.2 ± 1.30 ^#^	1.54 ± 0.09	5.54 ± 0.27
Rottlerin	26.3 ± 0.98 ^#^	9.32 ± 0.25 ^#^	13.3 ± 0.68 ^#^	1.68 ± 0.05	5.08 ± 0.13

Mean ± S.E.M. *n* = 7–10. * *p* < 0.05 vs. control in N-IND for each category. ^#^ *p* < 0.05 vs. control in N-IND/MEN for each category.

## Data Availability

Not applicable.

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
