# Peer review of "Energy-Dependent Endocytosis Is Responsible for Skin Penetration of Formulations Based on a Combination of Indomethacin Nanoparticles and l-Menthol in Rat and Göttingen Minipig"

_ijms, 2021, doi:10.3390/ijms22105137_

Round 1

Reviewer 1 Report

Hiroko Otake et al evaluated the penetration of Indomethacin nanoparticles with or with endocytosis inhibitors in rat and minipig skins.

  1. How the authors chose the concentrations of endocytosis inhibitors for skin penetration studies? The authors need to mention the experimental data or literature explaining the concentrations in the manuscript.
  2. In Table 1, mention the mean±SD/SEM for the footnote.
  3. Line 113, 116: the values mentioned are not matching with the figures. The authors need to modify the values.
  4. Y-axis of all figures represents cumulative IND concentration not IND concentration. It needs to be modified for A and C in all figures.
  5. Why the skin penetration experiments were performed for 24 hrs? Did authors check the viability, tissue integrity, tight junctions and TEER values of skin tissues for 24 hrs duration? Did authors perform permeability markers for skin penetration studies for 24 hrs? The authors should mention this information in the manuscript.
  6. Tables 2 and 3 represent the penetration/permeability parameters not the pharmacokinetic parameters. Modify the information in the table legends and text.
  7. In Tables 2 and 3, the unit mentioned for Jc is in nmol/cm2/h. Is it represents nmol or umol? Check the units and correct them.
  8. Why the authors didn’t perform the in vivo pharmacokinetics studies of IND formulations with and without endocytosis inhibitors using normal skin and SC removed skin in rats? This information would be useful for ex vivo and in vivo correlation.
  9. What would be the effect of the IND formulation and inhibitors on the tissue integrity of the skin?
  10. The authors should include the histology data of pre- and post- treated skin tissues of rats and minipigs with and without inhibitors in the manuscript.
  11. Line 277, include the weight of the rats used for these studies.
  12. Line 278, include the age and sex of the minipigs used in these studies.
  13. Line 292, rewrite the sentence.
  14. Line 298, remove Japan after sigma-Aldrich.
  15. Line 324, correct the word trenadermal.
  16. Line 359, mention the mean±SD/SEM for the thickness of skin tissues used for both rat and minipig.   

Author Response

We carefully revised our manuscript according to the suggestions of the reviewer 1, and details are as follows.

< Q and A for Reviewer 1>

Q1. How the authors chose the concentrations of endocytosis inhibitors for skin penetration studies? The authors need to mention the experimental data or literature explaining the concentrations in the manuscript.

A1. The reviewer’s comment is correct. We determined the endocytosis inhibitors and its concentration according to previous reports (Ref. 25-27). In order to respond to the reviewer’s comment, we mentioned the information and references in the Materials and Methods (line 363-364, Reference 25-27).

Ref. 25.  Malomouzh et al. The effect of dynasore, a blocker of dy-namin-dependent endocytosis, on spontaneous quantal and non-quantal release of acetylcholine in murine neuromuscular junctions. Dokl. Biol. Sci. 2014, 459, 330–333, doi: 10.1134/S0012496614060052.

Ref. 26.  Mäger et al. The role of endocytosis on the uptake kinetics of lucifer-in-conjugated cell-penetrating peptides. Biochim. Biophys. Acta 2012, 1818, 502–511, doi: 10.1016/j.bbamem.2011.11.020.

Ref. 27.  Hufnagel et al. Fluid phase endocytosis contributes to transfection of DNA by PEI-25. Mol. Ther. 2009, 17, 1411–1417, doi: 10.1038/mt.2009.121.

Q2. In Table 1, mention the mean±SD/SEM for the footnote.

A2. In order to respond to the reviewer’s comment, we mentioned the mean±S.E.M. for the footnote (Table 1 legend).

Q3. Line 113, 116: the values mentioned are not matching with the figures. The authors need to modify the values.

A3. The reviewer’s comments are very important. In order to respond to the reviewer’s comment, we collected the values (line 113, 116).

Q4. Y-axis of all figures represents cumulative IND concentration not IND concentration. It needs to be modified for A and C in all figures.

A4. The reviewer’s comment is correct. In order to respond to the reviewer’s comment, we revised to “cumulative IND concentration” from “IND concentration” (Figure 1-4).

Q5. Why the skin penetration experiments were performed for 24 hrs? Did authors check the viability, tissue integrity, tight junctions and TEER values of skin tissues for 24 hrs duration? Did authors perform permeability markers for skin penetration studies for 24 hrs? The authors should mention this information in the manuscript.

A5. Thank you for pointing out this. Many reports containing our previous studies (Ref. 10, 11 and 13) have showed that the integrity of skin tissue was preserved for 24 h in the in vitro skin penetration experiments using rat. Taken together, we determined the measurement time in the study. In addition, in the in vitro skin penetration experiments, the rapid increase (burst) in drug penetration was observed when the skin tissue was broken (decrease of TEER), although the drug penetration was linearity for 24 h in this study. These results suggested the measurement time is not problem in the skin experiments. On the other hand, it is important to check the tissue integrity. Therefore, we added the necessity of histology data in the Discussion. Thank you very much for pointing this out (line 220-225, Reference 10, 11 and 13).

Ref. 10.  Nagai et al. Design of a transdermal formulation containing raloxifene na-noparticles for osteoporosis treatment. Int. J. Nanomedicine 2018, 13, 5215-5229.

Ref. 11.  Nagai et al. Involvement of Endocytosis in the Transdermal Penetration Mechanism of Ketoprofen Nanoparticles. Int. J. Mol. Sci. 2018, 19, 2138.

Ref. 13.  Nagai et al. Combination with l-Menthol En-hances Transdermal Penetration of Indomethacin Solid Nanoparticles. Int. J. Mol. Sci. 2019, 20, 3644.

Q6. Tables 2 and 3 represent the penetration/permeability parameters not the pharmacokinetic parameters. Modify the information in the table legends and text.

A6. The reviewer’s comment is correct. In order to respond to the reviewer’s comment, we revised to “penetration/permeability parameters” from “pharmacokinetic parameters” in the Table legends and text (line 152, 160, Table 2 and 3 legend).

Q7. In Tables 2 and 3, the unit mentioned for Jc is in nmol/cm2/h. Is it represents nmol or umol? Check the units and correct them.

A7. Thank you for pointing out this. In order to respond to the reviewer’s comment, we checked the units, and it is correct. Thank you very much for pointing this out.

Q8. Why the authors didn’t perform the in vivo pharmacokinetics studies of IND formulations with and without endocytosis inhibitors using normal skin and SC removed skin in rats? This information would be useful for ex vivo and in vivo correlation.

A8. The reviewer’s comments are very important. It was known that the structure and penetration of Göttingen minipig skin were closely similar to those of human skin, and the Göttingen minipigs have been used to predict human pharmacokinetics profiles. Taken together, we attempted to investigate the effect of l-menthol on SC penetration of IND-NPs, and demonstrated that these forms of endocytosis (CavME, CME, and MP) are related to the skin penetration of IND from transdermal gel based on IND-NPs by in vitro study using both of rat and Göttingen minipig skin. From these purpose, the in vitro study using rat and Göttingen minipig skin were performed in this study. On the other hand, the in vivo pharmacokinetics studies is useful for ex vivo and in vivo correlation. Therefore, we added the importance of in vivo pharmacokinetics studies in the discussion. Thank you very much for pointing this out (line 286-288).

Q9. What would be the effect of the IND formulation and inhibitors on the tissue integrity of the skin?

A9. Thank you for pointing out this. Our previous reports using rat skin showed that the 0.5% DMSO (vehicle) was not affect the skin penetration, and the AUCSkin in vehicle-treated skin was similar to normal-skin (Ref. 10). In addition, the no rapid increase (burst) in IND permeation was observed during the measurement period (0-24 h), and S.E.M. values were not big in this study (Fig. 3 and 4). These results suggested that the integrity of skin tissue treated with IND formulation and inhibitor was not problem in the in vitro skin penetration study. In order to respond to the reviewer’s comment, we add these contents (line 220-225, reference 10).

Ref. 10.  Nagai et al. Design of a transdermal formulation containing raloxifene na-noparticles for osteoporosis treatment. Int. J. Nanomedicine 2018, 13, 5215-5229, doi: 10.2147/IJN.S173216.

Q10. The authors should include the histology data of pre- and post- treated skin tissues of rats and minipigs with and without inhibitors in the manuscript.

A10. Thank you very much for pointing this out. It is important the histology data to evaluate the tissue integrity of the skin, although we don’t have the histology data in this study. On the other hand, overlapping with the answer in Q9, our previous reports using rat skin showed that the 0.5% DMSO (vehicle) was not affect the skin penetration, and the AUCSkin in vehicle-treated skin was similar to normal-skin (Ref. 10). In addition, the no rapid increase (burst) in IND permeation was observed during the measurement period (0-24 h), and S.E.M. values were not big in this study (Fig. 3 and 4). These results suggested that the integrity of skin tissue treated with IND formulation and inhibitor was not problem in the in vitro skin penetration study. We added the contents, and mentioned the importance of histology data. Thank you for pointing out this (line 220-225, 285-288).

Q11. Line 277, include the weight of the rats used for these studies.

A11. The reviewer’s comments are very important. The weight of the rats were 281±8 g (n=82, mean±S.E.M.). In order to respond to the reviewer’s comment, we added these contents in the Materials and Methods (line 291).

Q12. Line 278, include the age and sex of the minipigs used in these studies.

A12. Thank you very much for pointing this out. The male Göttingen minipig, age 5 months, were used in this study. In order to respond to the reviewer’s comment, we added these contents in the Materials and Methods (line 292).

Q13. Line 292, rewrite the sentence.

Line 298, remove Japan after sigma-Aldrich.

Line 324, correct the word trenadermal.

Line 359, mention the mean±SD/SEM for the thickness of skin tissues used for both rat and minipig.

A13. The reviewer’s comment is correct. The thicknesses of the skins of rats and Göttingen minipigs are 0.086±0.011 cm and 0.133±0.021 cm, respectively (n=10, mean±S.E.M). In order to respond to the reviewer’s comment, we corrected these sentences (line 307, 313, 341, 380-381).

Thank you for great comments.

Reviewer 2 Report

I feel that the authors provided an interesting and well-done study. I would accept it. 

Author Response

We carefully revised our manuscript according to the suggestions of the reviewer 2, and details are as follows.

< Q and A for Reviewer 2>

Q1. I feel that the authors provided an interesting and well-done study. I would accept it.

A1. Thank you very much for great comments.

Reviewer 3 Report

English of the manuscript can be improved. The manuscript is not written smoothly probably due to the lack of certain transitions in the content. 

Below are some other problems:

1,"and the soluble IND levels were determined as soluble. " Is it missing something here? It's difficult to understand.

2, How to determine the drugs in the skin? The preparation protocol is missing in the method section.

3, Why did the authors select the used inhibitor concentrations?

4, The authors should report the detection limit and linearity range of their HPLC method.

Author Response

We carefully revised our manuscript according to the suggestions of the reviewer 3, and details are as follows.

< Q and A for Reviewer 3>

Q1. English of the manuscript can be improved. The manuscript is not written smoothly probably due to the lack of certain transitions in the content.

A1. Thank you very much for pointing this out. In order to respond to the reviewer’s comment, the manuscript was checked and edited by a native English-speaking person with sufficient scientific knowledge (MDPI English editing service, English editing ID: English-29714).

Q2. "and the soluble IND levels were determined as soluble. "Is it missing something here? It's difficult to understand.

A2. The reviewer’s comment is correct. In order to respond to the reviewer’s comment, we revised this sentence to “The soluble IND and solid IND (IND-NPs) in the transdermal formulations were separated by centrifugation (1×105 g) using an OptimaTM MAX-XP Ultracentrifuge (Beckman Coulter, Osaka, Japan), and the levels of soluble IND and IND-NPs were analyzed by the HPLC method described below” (line 338-343).

Q3. How to determine the drugs in the skin? The preparation protocol is missing in the method section.

A3. Thank you very much for pointing this out. The IND in the samples was extracted by methanol, and the IND levels were analyzed by the HPLC method. In order to respond to the reviewer’s comment, we added the content in the Materials and Methods (line 361-362).

Q4. Why did the authors select the used inhibitor concentrations?

A4. The reviewer’s comments are very important. We determined the endocytosis inhibitors and its concentration according to previous reports (Ref. 25-27). In order to respond to the reviewer’s comment, we mentioned the information and references in the Materials and Methods (line 363-364, Reference 25-27).

Ref. 25.  Malomouzh et al. The effect of dynasore, a blocker of dy-namin-dependent endocytosis, on spontaneous quantal and non-quantal release of acetylcholine in murine neuromuscular junctions. Dokl. Biol. Sci. 2014, 459, 330–333, doi: 10.1134/S0012496614060052.

Ref. 26.  Mäger et al. The role of endocytosis on the uptake kinetics of lucifer-in-conjugated cell-penetrating peptides. Biochim. Biophys. Acta 2012, 1818, 502–511, doi: 10.1016/j.bbamem.2011.11.020.

Ref. 27.  Hufnagel et al. Fluid phase endocytosis contributes to transfection of DNA by PEI-25. Mol. Ther. 2009, 17, 1411–1417, doi: 10.1038/mt.2009.121.

Q5. The authors should report the detection limit and linearity range of their HPLC method.

A5. Thank you for pointing out this. We used the propyl p-hydroxybenzoate as the internal standard, and calibration curve was created with the concentration of IND set to 0.1-10 µg/mL (0.28-28 µM). The calibration curve was linear (y=0.2668x-0.0026, r=0.9999), and the detection limit was approximately 0.1 µg/mL (0.28 µM). In order to respond to the reviewer’s comment, we added the contents in the Materials and Methods (line 351-352).

Thank you for great comments.

Round 2

Reviewer 1 Report

Hiroko Otake et al significantly improved the manuscript and answered all the comments.

In figures 1, 2 and 3, tick marks should be outside the axis. These tick marks overlap with the data points.  

Author Response

< Q and A for Reviewer 1>

Q1. In figures 1, 2 and 3, tick marks should be outside the axis. These tick marks overlap with the data points. 

A1. The reviewer’s comment is correct. In order to respond to the reviewer’s comment, we revised that tick marks should be outside the axis. Thank you very much for pointing this out (Figures 1-4).

Thank you for great comments.

Reviewer 3 Report

No more comments.

Author Response

< Q and A for Reviewer 3>

Q1. No more comments.

A1. Thank you very much for great review.
